# BACKSHIFT: Learning causal cyclic graphs from unknown shift interventions

**Dominik Rothenhäusler**[*]
Seminar für Statistik
ETH Zürich, Switzerland
rothenhaeusler@stat.math.ethz.ch

**Christina Heinze**[*]
Seminar für Statistik
ETH Zürich, Switzerland
heinze@stat.math.ethz.ch

**Jonas Peters**
Max Planck Institute for Intelligent Systems
Tübingen, Germany
jonas.peters@tuebingen.mpg.de

**Nicolai Meinshausen**
Seminar für Statistik
ETH Zürich, Switzerland
meinshausen@stat.math.ethz.ch

## Abstract

We propose a simple method to learn linear causal cyclic models in the presence of latent variables. The method relies on equilibrium data of the model recorded under a specific kind of interventions ("shift interventions"). The location and strength of these interventions do not have to be known and can be estimated from the data. Our method, called BACKSHIFT, only uses second moments of the data and performs simple joint matrix diagonalization, applied to differences between covariance matrices. We give a sufficient and necessary condition for identifiability of the system, which is fulfilled almost surely under some quite general assumptions if and only if there are at least three distinct experimental settings, one of which can be pure observational data. We demonstrate the performance on some simulated data and applications in flow cytometry and financial time series.

## 1 Introduction

Discovering causal effects is a fundamentally important yet very challenging task in various disciplines, from public health research and sociological studies, economics to many applications in the life sciences. There has been much progress on learning acyclic graphs in the context of structural equation models [1], including methods that learn from observational data alone under a faithfulness assumption [2, 3, 4, 5], exploiting non-Gaussianity of the data [6, 7] or non-linearities [8]. Feedbacks are prevalent in most applications, and we are interested in the setting of [9], where we observe the equilibrium data of a model that is characterized by a set of linear relations

$$\mathbf{x} = \mathbf{B}\mathbf{x} + \mathbf{e}, \tag{1}$$

where $\mathbf{x} \in \mathbb{R}^p$ is a random vector and $\mathbf{B} \in \mathbb{R}^{p \times p}$ is the connectivity matrix with zeros on the diagonal (no self-loops). Allowing for self-loops would lead to an identifiability problem, independent of the method. See Section B in the Appendix for more details on this setting. The graph corresponding to $\mathbf{B}$ has $p$ nodes and an edge from node $j$ to node $i$ if and only if $\mathbf{B}_{i,j} \neq 0$. The error terms $\mathbf{e}$ are $p$-dimensional random variables with mean 0 and positive semi-definite covariance matrix $\mathbf{\Sigma_e} = E(\mathbf{e}\mathbf{e}^T)$. We do not assume that $\mathbf{\Sigma_e}$ is a diagonal matrix which allows the existence of latent variables.

The solutions to (1) can be thought of as the deterministic equilibrium solutions (conditional on the noise term) of a dynamic model governed by first-order difference equations with matrix $\mathbf{B}$ in the

---

[*]Authors contributed equally.

sense of [10]. For well-defined equilibrium solutions of (1), we need that $\mathbf{I} - \mathbf{B}$ is invertible. Usually we also want (1) to converge to an equilibrium when iterating as $\mathbf{x}^{(new)} \leftarrow \mathbf{B}\mathbf{x}^{(old)} + \mathbf{e}$ or in other words $\lim_{m \to \infty} \mathbf{B}^m \equiv \mathbf{0}$. This condition is equivalent to the spectral radius of $\mathbf{B}$ being strictly smaller than one [11]. We will make an assumption on cyclic graphs that restricts the strength of the feedback. Specifically, let a cycle of length $\eta$ be given by $(m_1, \ldots, m_{\eta+1} = m_1) \in \{1, \ldots, p\}^{1+\eta}$ and $m_k \neq m_\ell$ for $1 \le k < \ell \le \eta$. We define the cycle-product $CP(\mathbf{B})$ of a matrix $\mathbf{B}$ to be the maximum over cycles of all lengths $1 < \eta \le p$ of the path-products

$$CP(\mathbf{B}) := \max_{\substack{(m_1, \ldots, m_\eta, m_{\eta+1}) \text{ cycle} \\ 1 < \eta \le p}} \prod_{1 \le k \le \eta} \left| \mathbf{B}_{m_{k+1}, m_k} \right|. \tag{2}$$

The cycle-product $CP(\mathbf{B})$ is clearly zero for acyclic graphs. We will assume the cycle-product to be strictly smaller than one for identifiability results, see Assumption (A) below. The most interesting graphs are those for which $CP(\mathbf{B}) < 1$ and for which the spectral radius of $\mathbf{B}$ is strictly smaller than one. Note that these two conditions are identical as long as the cycles in the graph do not intersect, i.e., there is no node that is part of two cycles (for example if there is at most one cycle in the graph). If cycles do intersect, we can have models for which either (i) $CP(\mathbf{B}) < 1$ but the spectral radius is larger than one or (ii) $CP(\mathbf{B}) > 1$ but the spectral radius is strictly smaller than one. Models in situation (ii) are not stable in the sense that the iterations will not converge under interventions. We can for example block all but one cycle. If this one single unblocked cycle has a cycle-product larger than 1 (and there is such a cycle in the graph if $CP(\mathbf{B}) > 1$), then the solutions of the iteration are not stable[1]. Models in situation (i) are not stable either, even in the absence of interventions. We can still in theory obtain the now instable equilibrium solutions to (1) as $(\mathbf{I} - \mathbf{B})^{-1}\mathbf{e}$ and the theory below applies to these instable equilibrium solutions. However, such instable equilibrium solutions are arguably of little practical interest. In summary: all interesting feedback models that are stable under interventions satisfy both $CP(\mathbf{B}) < 1$ and have a spectral radius strictly smaller than one. We will just assume $CP(\mathbf{B}) < 1$ for the following results.

It is impossible to learn the structure $\mathbf{B}$ of this model from observational data alone without making further assumptions. The LINGAM approach has been extended in [11] to cyclic models, exploiting a possible non-Gaussianity of the data. Using both experimental and interventional data, [12, 9] could show identifiability of the connectivity matrix $\mathbf{B}$ under a learning mechanism that relies on data under so-called "surgical" or "perfect" interventions. In their framework, a variable becomes independent of all its parents if it is being intervened on and all incoming contributions are thus effectively removed under the intervention (also called do-interventions in the classical sense of [13]). The learning mechanism makes then use of the knowledge where these "surgical" interventions occurred. [14] also allow for "changing" the incoming arrows for variables that are intervened on; but again, [14] requires the location of the interventions while we do not assume such knowledge. [15] consider a target variable and allow for arbitrary interventions on all other nodes. They neither permit hidden variables nor cycles.

Here, we are interested in a setting where we have either no or just very limited knowledge about the exact location and strength of the interventions, as is often the case for data observed under different environments (see the example on financial time series further below) or for biological data [16, 17]. These interventions have been called "fat-hand" or "uncertain" interventions [18]. While [18] assume acyclicity and model the structure explicitly in a Bayesian setting, we assume that the data in environment $j$ are equilibrium observations of the model

$$\mathbf{x}_j = \mathbf{B}\mathbf{x}_j + \mathbf{c}_j + \mathbf{e}_j, \tag{3}$$

where the random intervention shift $\mathbf{c}_j$ has a mean and covariance $\boldsymbol{\Sigma}_{\mathbf{c},j}$. The *location* of these interventions (or simply the *intervened variables*) are those components of $\mathbf{c}_j$ that are not zero with probability one. Given these locations, the interventions simply shift the variables by a value determined by $\mathbf{c}_j$; they are therefore not "surgical" but can be seen as a special case of what is called an "imperfect", "parametric" [19] or "dependent" intervention [20] or "mechanism change" [21]. The matrix $\mathbf{B}$ and the error distribution of $\mathbf{e}_j$ are assumed to be identical in all environments. In contrast to the covariance matrix for the noise term $\mathbf{e}_j$, we *do* assume that $\boldsymbol{\Sigma}_{\mathbf{c},j}$ is a diagonal

matrix, which is equivalent to demanding that interventions at different variables are uncorrelated. This is a key assumption necessary to identify the model using experimental data. Furthermore, we will discuss in Section 4.2 how a violation of the model assumption (3) can be detected and used to estimate the location of the interventions.

In Section 2 we show how to leverage observations under different environments with different interventional distributions to learn the structure of the connectivity matrix $\mathbf{B}$ in model (3). The method rests on a simple joint matrix diagonalization. We will prove necessary and sufficient conditions for identifiability in Section 3. Numerical results for simulated data and applications in flow cytometry and financial data are shown in Section 4.

## 2 Method

### 2.1 Grouping of data

Let $\mathcal{J}$ be the set of experimental conditions under which we observe equilibrium data from model (3). These different experimental conditions can arise in two ways: (a) a controlled experiment was conducted where the external input or the external imperfect interventions have been deliberately changed from one member of $\mathcal{J}$ to the next. An example are the flow cytometry data [22] discussed in Section 4.2. (b) The data are recorded over time. It is assumed that the external input is changing over time but not in an explicitly controlled way. The data are grouped into consecutive blocks $j \in \mathcal{J}$ of observations, see Section 4.3 for an example.

### 2.2 Notation

Assume we have $n_j$ observations in each setting $j \in \mathcal{J}$. Let $\mathbf{X}_j$ be the $(n_j \times p)$-matrix of observations from model (3). For general random variables $\mathbf{a}_j \in \mathbb{R}^p$, the population covariance matrix in setting $j \in \mathcal{J}$ is called $\mathbf{\Sigma}_{\mathbf{a},j} = \text{Cov}(\mathbf{a}_j)$, where the covariance is under the setting $j \in \mathcal{J}$. Furthermore, the covariance on all settings except setting $j \in \mathcal{J}$ is defined as an average over all environments except for the $j$-th environment, $(|\mathcal{J}|-1)\mathbf{\Sigma}_{\mathbf{c},-j} := \sum_{j' \in \mathcal{J} \setminus \{j\}} \mathbf{\Sigma}_{\mathbf{c},j'}$. The population Gram matrix is defined as $\mathbf{G}_{\mathbf{a},j} = E(\mathbf{a}_j \mathbf{a}_j{}^T)$. Let the $(p \times p)$-dimensional $\hat{\mathbf{\Sigma}}_{\mathbf{a},j}$ be the empirical covariance matrix of the observations $\mathbf{A}_j \in \mathbb{R}^{n_j \times p}$ of variable $\mathbf{a}_j$ in setting $j \in \mathcal{J}$. More precisely, let $\tilde{\mathbf{A}}_j$ be the column-wise mean-centered version of $\mathbf{A}_j$. Then $\hat{\mathbf{\Sigma}}_{\mathbf{a},j} := (n_j - 1)^{-1} \tilde{\mathbf{A}}_j^T \tilde{\mathbf{A}}_j$. The empirical Gram matrix is denoted by $\hat{\mathbf{G}}_{\mathbf{a},j} := n_j^{-1} \mathbf{A}_j^T \mathbf{A}_j$.

### 2.3 Assumptions

The main assumptions have been stated already but we give a summary below.

(A) The data are observations of the equilibrium observations of model (3). The matrix $\mathbf{I} - \mathbf{B}$ is invertible and the solutions to (3) are thus well defined. The cycle-product (2) $CP(\mathbf{B})$ is strictly smaller than one. The diagonal entries of $\mathbf{B}$ are zero.

(B) The distribution of the noise $\mathbf{e}_j$ (which includes the influence of latent variables) and the connectivity matrix $\mathbf{B}$ are identical across all settings $j \in \mathcal{J}$. In each setting $j \in \mathcal{J}$, the intervention shift $\mathbf{c}_j$ and the noise $\mathbf{e}_j$ are uncorrelated.

(C) Interventions at different variables in the same setting are uncorrelated, that is $\mathbf{\Sigma}_{\mathbf{c},j}$ is an (unknown) diagonal matrix for all $j \in \mathcal{J}$.

We will discuss a stricter version of (C) in Section D in the Appendix that allows the use of Gram matrices instead of covariance matrices. The conditions above imply that the environments are characterized by different interventions strength, as measured by the variance of the shift $\mathbf{c}$ in each setting. We aim to reconstruct both the connectivity matrix $\mathbf{B}$ from observations in different environments and also aim to reconstruct the a-priori unknown intervention strength and location in each environment. Additionally, we will show examples where we can detect violations of the model assumptions and use these to reconstruct the location of interventions.

### 2.4 Population method

The main idea is very simple. Looking at the model (3), we can rewrite

$$(\mathbf{I} - \mathbf{B})\mathbf{x}_j = \mathbf{c}_j + \mathbf{e}_j. \tag{4}$$

The population covariance of the transformed observations are then for all settings $j \in \mathcal{J}$ given by

$$(\mathbf{I} - \mathbf{B})\boldsymbol{\Sigma}_{\mathbf{x},j}(\mathbf{I} - \mathbf{B})^T = \boldsymbol{\Sigma}_{\mathbf{c},j} + \boldsymbol{\Sigma}_{\mathbf{e}}. \tag{5}$$

The last term $\boldsymbol{\Sigma}_{\mathbf{e}}$ is constant across all settings $j \in \mathcal{J}$ (but not necessarily diagonal as we allow hidden variables). Any change of the matrix on the left-hand side thus stems from a shift in the covariance matrix $\boldsymbol{\Sigma}_{\mathbf{c},j}$ of the interventions. Let us define the difference between the covariance of $\mathbf{c}$ and $\mathbf{x}$ in setting $j$ as

$$\boldsymbol{\Delta}\boldsymbol{\Sigma}_{\mathbf{c},j} := \boldsymbol{\Sigma}_{\mathbf{c},j} - \boldsymbol{\Sigma}_{\mathbf{c},-j}, \quad \text{and} \quad \boldsymbol{\Delta}\boldsymbol{\Sigma}_{\mathbf{x},j} := \boldsymbol{\Sigma}_{\mathbf{x},j} - \boldsymbol{\Sigma}_{\mathbf{x},-j}. \tag{6}$$

Assumption (B) together with (5) implies that

$$(\mathbf{I} - \mathbf{B})\boldsymbol{\Delta}\boldsymbol{\Sigma}_{\mathbf{x},j}(\mathbf{I} - \mathbf{B})^T = \boldsymbol{\Delta}\boldsymbol{\Sigma}_{\mathbf{c},j} \qquad \forall j \in \mathcal{J}. \tag{7}$$

Using assumption (C), the random intervention shifts at different variables are uncorrelated and the right-hand side in (7) is thus a diagonal matrix for all $j \in \mathcal{J}$. Let $\mathcal{D} \subset \mathbb{R}^{p \times p}$ be the set of all invertible matrices. We also define a more restricted space $\mathcal{D}_{cp}$ which only includes those members of $\mathcal{D}$ that have entries all equal to one on the diagonal and have a cycle-product less than one,

$$\mathcal{D} := \left\{ \mathbf{D} \in \mathbb{R}^{p \times p} : \mathbf{D} \text{ invertible} \right\} \tag{8}$$

$$\mathcal{D}_{cp} := \left\{ \mathbf{D} \in \mathbb{R}^{p \times p} : \mathbf{D} \in \mathcal{D} \text{ and } \operatorname{diag}(\mathbf{D}) \equiv 1 \text{ and } CP(\mathbf{I} - \mathbf{D}) < 1 \right\}. \tag{9}$$

Under Assumption (A), $\mathbf{I} - \mathbf{B} \in \mathcal{D}_{cp}$. Motivated by (7), we now consider the minimizer

$$\mathbf{D} = \operatorname{argmin}_{\mathbf{D}' \in \mathcal{D}_{cp}} \sum_{j \in \mathcal{J}} L(\mathbf{D}' \boldsymbol{\Delta}\boldsymbol{\Sigma}_{\mathbf{x},j} \mathbf{D}'^T), \quad \text{where } L(\mathbf{A}) := \sum_{k \neq l} \mathbf{A}_{k,l}^2 \tag{10}$$

is the loss $L$ for any matrix $\mathbf{A}$ and defined as the sum of the squared off-diagonal elements. In Section 3, we present necessary and sufficient conditions on the interventions under which $\mathbf{D} = \mathbf{I} - \mathbf{B}$ is the unique minimizer of (10). In this case, exact joint diagonalization is possible so that $L(\mathbf{D}\boldsymbol{\Delta}\boldsymbol{\Sigma}_{\mathbf{x},j}\mathbf{D}^T) = 0$ for all environments $j \in \mathcal{J}$. We discuss an alternative that replaces covariance with Gram matrices throughout in Section D in the Appendix. We now give a finite-sample version.

## 2.5 Finite-sample estimate of the connectivity matrix

In practice, we estimate $\mathbf{B}$ by minimizing the empirical counterpart of (10) in two steps. First, the solution of the optimization is only constrained to matrices in $\mathcal{D}$. Subsequently, we enforce the constraint on the solution to be a member of $\mathcal{D}_{cp}$. The BACKSHIFT algorithm is presented in Algorithm 1 and we describe the important steps in more detail below.

---
**Algorithm 1** BACKSHIFT
---
**Input:** $\mathbf{X}_j \ \forall j \in \mathcal{J}$
  1: Compute $\widehat{\boldsymbol{\Delta}\boldsymbol{\Sigma}}_{\mathbf{x},j} \ \forall j \in \mathcal{J}$
  2: $\tilde{\mathbf{D}} = \text{FFDIAG}(\widehat{\boldsymbol{\Delta}\boldsymbol{\Sigma}}_{\mathbf{x},j})$
  3: $\hat{\mathbf{D}} = \texttt{PermuteAndScale}(\tilde{\mathbf{D}})$
  4: $\hat{\mathbf{B}} = \mathbf{I} - \hat{\mathbf{D}}$
**Output:** $\hat{\mathbf{B}}$
---

**Steps 1 & 2.** First, we minimize the following empirical, less constrained variant of (10)

$$\tilde{\mathbf{D}} := \operatorname{argmin}_{\mathbf{D}' \in \mathcal{D}} \sum_{j \in \mathcal{J}} L(\mathbf{D}'(\widehat{\boldsymbol{\Delta}\boldsymbol{\Sigma}}_{\mathbf{x},j})\mathbf{D}'^T), \tag{11}$$

where the population differences between covariance matrices are replaced with their empirical counterparts and the only constraint on the solution is that it is invertible, i.e. $\tilde{\mathbf{D}} \in \mathcal{D}$. For the optimization we use the joint approximate matrix diagonalization algorithm FFDIAG [23].

**Step 3.** The constraint on the cycle product and the diagonal elements of $\mathbf{D}$ is enforced by (a) permuting and (b) scaling the rows of $\tilde{\mathbf{D}}$. Part (b) simply scales the rows so that the diagonal elements of the resulting matrix $\hat{\mathbf{D}}$ are all equal to one. The more challenging first step (a) consists of finding a permutation such that under this permutation the scaled matrix from part (b) will have a cycle product as small as possible (as follows from Theorem 3, at most one permutation can lead to a cycle product less than one). This optimization problem seems computationally challenging at first, but we show that it can be solved by a variant of the *linear assignment problem* (LAP) (see e.g. [24]), as proven in Theorem 3 in the Appendix. As a last step, we check whether the cycle product of $\hat{\mathbf{D}}$ is less than one, in which case we have found the solution. Otherwise, no solution satisfying the model assumptions exists and we return a warning that the model assumptions are not met. See Appendix B for more details.

**Computational cost.** The computational complexity of BACKSHIFT is $O(|\mathcal{J}| \cdot n \cdot p^2)$ as computing the covariance matrices costs $O(|\mathcal{J}| \cdot n \cdot p^2)$, FFDIAG has a computational cost of $O(|\mathcal{J}| \cdot p^2)$ and both the linear assignment problem and computing the cycle product can be solved in $O(p^3)$ time. For instance, this complexity is achieved when using the Hungarian algorithm for the linear assignment problem (see e.g. [24]) and the cycle product can be computed with a simple dynamic programming approach.

## 2.6 Estimating the intervention variances

One additional benefit of BACKSHIFT is that the location and strength of the interventions can be estimated from the data. The empirical, plug-in version of Eq. (7) is given by

$$(\mathbf{I} - \hat{\mathbf{B}})\widehat{\mathbf{\Delta\Sigma}}_{\mathbf{x},j}(\mathbf{I} - \hat{\mathbf{B}})^T = \widehat{\mathbf{\Delta\Sigma}}_{\mathbf{c},j} = \widehat{\mathbf{\Sigma}}_{\mathbf{c},j} - \widehat{\mathbf{\Sigma}}_{\mathbf{c},-j} \qquad \forall j \in \mathcal{J}. \tag{12}$$

So the element $(\widehat{\mathbf{\Delta\Sigma}}_{\mathbf{c},j})_{kk}$ is an estimate for the difference between the variance of the intervention at variable $k$ in environment $j$, namely $(\mathbf{\Sigma}_{\mathbf{c},j})_{kk}$, and the average in all other environments, $(\mathbf{\Sigma}_{\mathbf{c},-j})_{kk}$. From these differences we can compute the intervention variance for all environments up to an offset. By convention, we set the minimal intervention variance across all environments equal to zero. Alternatively, one can let observational data, if available, serve as a baseline against which the intervention variances are measured.

## 3 Identifiability

Let for simplicity of notation,
$$\boldsymbol{\eta}_{j,k} := (\mathbf{\Delta\Sigma}_{\mathbf{c},j})_{kk}$$
be the variance of the random intervention shifts $\mathbf{c}_j$ at node $k$ in environment $j \in \mathcal{J}$ as per the definition of $\mathbf{\Delta\Sigma}_{\mathbf{c},j}$ in (6). We then have the following identifiability result (the proof is provided in Appendix A).

**Theorem 1.** *Under assumptions (A), (B) and (C), the solution to* (10) *is unique if and only if for all* $k, l \in \{1, \ldots, p\}$ *there exist* $j, j' \in \mathcal{J}$ *such that*

$$\boldsymbol{\eta}_{j,k}\boldsymbol{\eta}_{j',l} \neq \boldsymbol{\eta}_{j,l}\boldsymbol{\eta}_{j',k}. \tag{13}$$

If none of the intervention variances $\boldsymbol{\eta}_{j,k}$ vanishes, the uniqueness condition is equivalent to demanding that the ratio between the intervention variances for two variables $k, l$ must not stay identical across all environments, that is there exist $j, j' \in \mathcal{J}$ such that

$$\frac{\boldsymbol{\eta}_{j,k}}{\boldsymbol{\eta}_{j,l}} \neq \frac{\boldsymbol{\eta}_{j',k}}{\boldsymbol{\eta}_{j',l}}, \tag{14}$$

which requires that the ratio of the variance of the intervention shifts at two nodes $k, l$ is not identical across all settings. This leads to the following corollary.

**Corollary 2.** *(i) The identifiability condition* (13) *cannot be satisfied if* $|\mathcal{J}| = 2$ *since then* $\boldsymbol{\eta}_{j,k} = -\boldsymbol{\eta}_{j',k}$ *for all* $k$ *and* $j \neq j'$. *We need at least three different environments for identifiability.*

*(ii) The identifiability condition* (13) *is satisfied for all* $|\mathcal{J}| \geq 3$ *almost surely if the variances of the intervention* $\mathbf{c}_j$ *are chosen independently (over all variables and environments* $j \in \mathcal{J}$) *from a distribution that is absolutely continuous with respect to Lebesgue measure.*

Condition (ii) can be relaxed but shows that we can already achieve full identifiability with a very generic setting for three (or more) different environments.

## 4 Numerical results

In this section, we present empirical results for both synthetic and real data sets. In addition to estimating the connectivity matrix $\mathbf{B}$, we demonstrate various ways to estimate properties of the interventions. Besides computing the point estimate for BACKSHIFT, we use *stability selection* [25] to assess the stability of retrieved edges. We attach R-code with which all simulations and analyses can be reproduced[2].

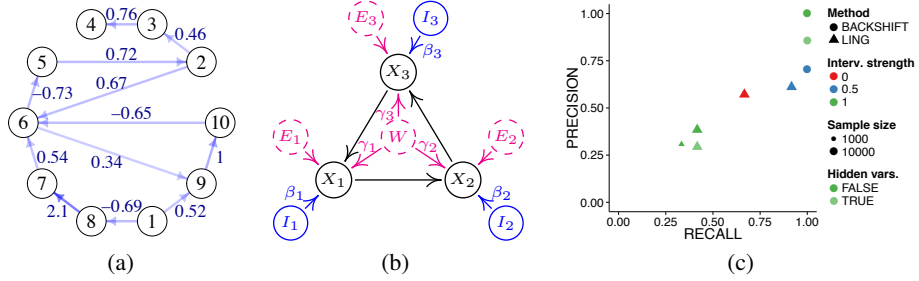

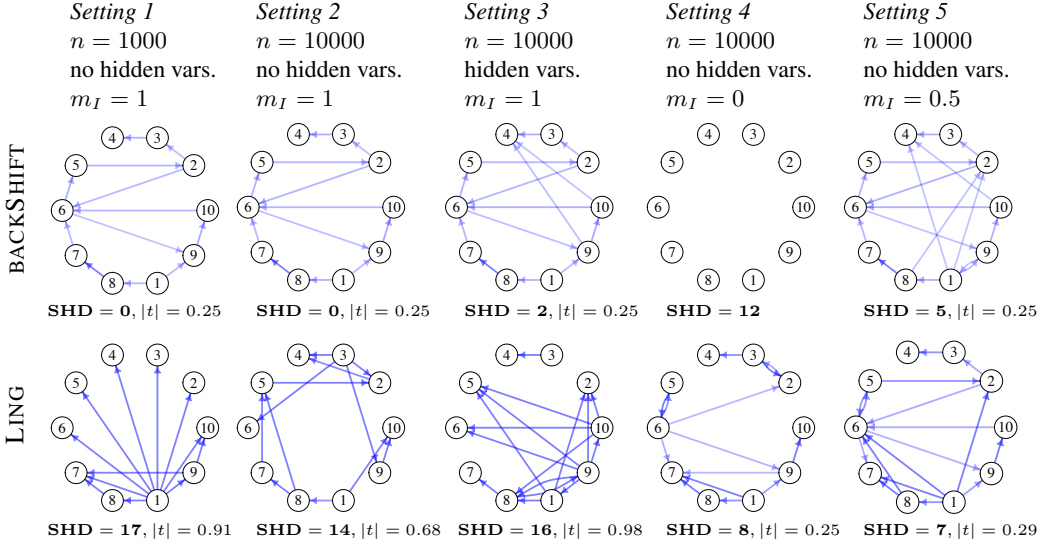

Figure 1: Simulated data. (a) True network. (b) Scheme for data generation. (c) Performance metrics for the settings considered in Section 4.1. For BACKSHIFT, precision and recall values for Settings 1 and 2 coincide.

Figure 2: Point estimates of BACKSHIFT and LING for synthetic data. We threshold the point estimate of BACKSHIFT at $t = \pm 0.25$ to exclude those entries which are close to zero. We then threshold the estimate of LING so that the two estimates have the same number of edges. In Setting 4, we threshold LING at $t = \pm 0.25$ as BACKSHIFT returns the empty graph. In Setting 3, it is not possible to achieve the same number of edges as all remaining coefficients in the point estimate of LING are equal to one in absolute value. The transparency of the edges illustrates the relative magnitude of the estimated coefficients. We report the structural Hamming distance (SHD) for each graph. Precision and recall values are shown in Figure 1(c).

## 4.1 Synthetic data

We compare the point estimate of BACKSHIFT against LING [11], a generalization of LINGAM to the cyclic case for purely observational data. We consider the cyclic graph shown in Figure 1(a) and generate data under different scenarios. The data generating mechanism is sketched in Figure 1(b). Specifically, we generate ten distinct environments with non-Gaussian noise. In each environment, the random intervention variable is generated as $(\mathbf{c}_j)_k = \beta_k^j I_k^j$, where $\beta_1^j, \ldots, \beta_p^j$ are drawn i.i.d. from $\text{Exp}(m_I)$ and $I_1^j, \ldots, I_p^j$ are independent standard normal random variables. The intervention shift thus acts on all observed random variables. The parameter $m_I$ regulates the strength of the intervention. If hidden variables exist, the noise term $(\mathbf{e}_j)_k$ of variable $k$ in environment $j$ is equal to $\gamma_k W^j$, where the weights $\gamma_1, \ldots, \gamma_p$ are sampled once from a $\mathcal{N}(0, 1)$-distribution and the random variable $W^j$ has a $\text{Laplace}(0, 1)$ distribution. If no hidden variables are present, then $(\mathbf{e}_j)_k$, $k = 1, \ldots, p$ is sampled i.i.d. $\text{Laplace}(0, 1)$. In this set of experiments, we consider five different settings (described below) in which the sample size $n$, the intervention strength $m_I$ as well as the existence of hidden variables varies.

We allow for hidden variables in only one out of five settings as LING assumes causal sufficiency and can thus in theory not cope with hidden variables. If no hidden variables are present, the pooled data can be interpreted as coming from a model whose error variables follow a mixture distribution. But if one of the error variables comes from the second mixture component, for example, the other

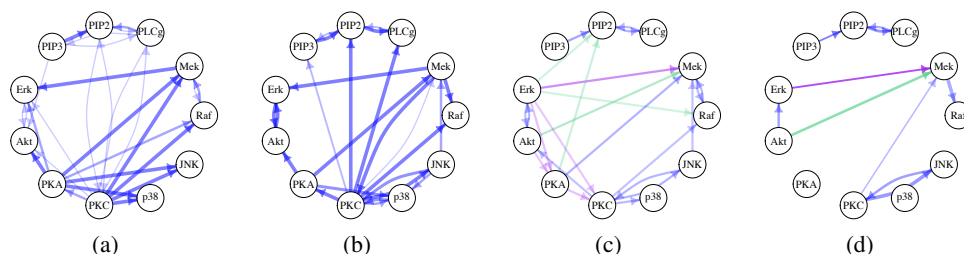

(a)                (b)                (c)                (d)

Figure 3: Flow cytometry data. (a) Union of the consensus network (according to [22]), the reconstruction by [22] and the best *acyclic* reconstruction by [26]. The edge thickness and intensity reflect in how many of these three sources that particular edge is present. (b) One of the *cyclic* reconstructions by [26]. The edge thickness and intensity reflect the probability of selecting that particular edge in the stability selection procedure. For more details see [26]. (c) BACKSHIFT point estimate, thresholded at $\pm 0.35$. The edge intensity reflects the relative magnitude of the coefficients and the coloring is a comparison to the union of the graphs shown in panels (a) and (b). Blue edges were also found in [26] and [22], purple edges are reversed and green edges were not previously found in (a) or (b). (d) BACKSHIFT stability selection result with parameters $\mathbb{E}(V) = 2$ and $\pi_{thr} = 0.75$. The edge thickness illustrates how often an edge was selected in the stability selection procedure.

error variables come from the second mixture component, too. In this sense, the data points are not independent anymore. This poses a challenge for LING which assumes an i.i.d. sample. We also cover a case (for $m_I = 0$) in which all assumptions of LING are satisfied (Scenario 4).

Figure 2 shows the estimated connectivity matrices for five different settings and Figure 1(c) shows the obtained precision and recall values. In Setting 1, $n = 1000$, $m_I = 1$ and there are no hidden variables. In Setting 2, $n$ is increased to $10000$ while the other parameters do not change. We observe that BACKSHIFT retrieves the correct adjacency matrix in both cases while LING's estimate is not very accurate. It improves slightly when increasing the sample size. In Setting 3, we do include hidden variables which violates the causal sufficiency assumption required for LING. Indeed, the estimate is worse than in Setting 2 but somewhat better than in Setting 1. BACKSHIFT retrieves two false positives in this case. Setting 4 is not feasible for BACKSHIFT as the distribution of the variables is identical in all environments (since $m_I = 0$). In Step 2 of the algorithm, FFDIAG does not converge and therefore the empty graph is returned. So the recall value is zero while precision is not defined. For LING all assumptions are satisfied and the estimate is more accurate than in the Settings 1–3. Lastly, Setting 5 shows that when increasing the intervention strength to $0.5$, BACKSHIFT returns a few false positives. Its performance is then similar to LING which returns its most accurate estimate in this scenario. The stability selection results for BACKSHIFT are provided in Figure 5 in Appendix E.

In short, these results suggest that the BACKSHIFT point estimates are close to the true graph if the interventions are sufficiently strong. Hidden variables make the estimation problem more difficult but the true graph is recovered if the strength of the intervention is increased (when increasing $m_I$ to $1.5$ in Setting 3, BACKSHIFT obtains a SHD of zero). In contrast, LING is unable to cope with hidden variables but also has worse accuracy in the absence of hidden variables under these shift interventions.

### 4.2 Flow cytometry data

The data published in [22] is an instance of a data set where the external interventions differ between the environments in $\mathcal{J}$ and might act on several compounds simultaneously [18]. There are nine different experimental conditions with each containing roughly 800 observations which correspond to measurements of the concentration of biochemical agents in single cells. The first setting corresponds to purely observational data.

In addition to the original work by [22], the data set has been described and analyzed in [18] and [26]. We compare against the results of [26], [22] and the "well-established consensus", according to [22], shown in Figures 3(a) and 3(b). Figure 3(c) shows the (thresholded) BACKSHIFT point estimate. Most of the retrieved edges were also found in at least one of the previous studies. Five edges are reversed in our estimate and three edges were not discovered previously. Figure 3(d) shows the corresponding stability selection result with the expected number of falsely selected variables

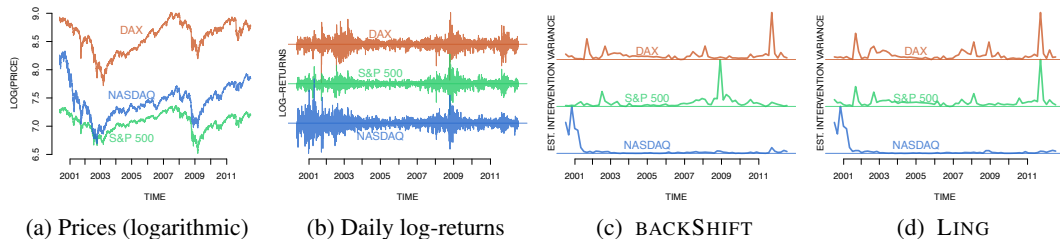

| (a) Prices (logarithmic) | (b) Daily log-returns | (c) BACKSHIFT | (d) LING |

Figure 4: Financial time series with three stock indices: NASDAQ (blue; technology index), S&P 500 (green; American equities) and DAX (red; German equities). (a) Prices of the three indices between May 2000 and end of 2011 on a logarithmic scale. (b) The scaled log-returns (daily change in log-price) of the three instruments are shown. Three periods of increased volatility are visible starting with the dot-com bust on the left to the financial crisis in 2008 and the August 2011 downturn. (c) The scaled estimated intervention variance with the estimated BACKSHIFT network. The three down-turns are clearly separated as originating in technology, American and European equities. (d) In contrast, the analogous LING estimated intervention variances have a peak in American equities intervention variance during the European debt crisis in 2011.

$\mathbb{E}(V) = 2$. This estimate is sparser in comparison to the other ones as it bounds the number of false discoveries. Notably, the feedback loops between PIP2 $\leftrightarrow$ PLCg and PKC $\leftrightarrow$ JNK were also found in [26].

It is also noteworthy that we can check the model assumptions of shift interventions, which is important for these data as they can be thought of as changing the mechanism or activity of a biochemical agent rather than regulate the biomarker directly [26]. If the shift interventions are not appropriate, we are in general not able to diagonalize the differences in the covariance matrices. Large off-diagonal elements in the estimate of the r.h.s in (7) indicate a mechanism change that is not just explained by a shift intervention as in (1). In four of the seven interventions environments with known intervention targets the largest mechanism violation happens directly at the presumed intervention target, see Appendix C for details. It is worth noting again that the presumed intervention target had not been used in reconstructing the network and mechanism violations.

### 4.3 Financial time series

Finally, we present an application in financial time series where the environment is clearly changing over time. We consider daily data from three stock indices NASDAQ, S&P 500 and DAX for a period between 2000-2012 and group the data into 74 overlapping blocks of 61 consecutive days each. We take log-returns, as shown in panel (b) of Figure 4 and estimate the connectivity matrix, which is fully connected in this case and perhaps of not so much interest in itself. It allows us, however, to estimate the intervention strength at each of the indices according to (12), shown in panel (c). The intervention variances separate very well the origins of the three major down-turns of the markets on the period. Technology is correctly estimated by BACKSHIFT to be at the epicenter of the dot-com crash in 2001 (NASDAQ as proxy), American equities during the financial crisis in 2008 (proxy is S&P 500) and European instruments (DAX as best proxy) during the August 2011 downturn.

## 5 Conclusion

We have shown that cyclic causal networks can be estimated if we obtain covariance matrices of the variables under unknown shift interventions in different environments. BACKSHIFT leverages solutions to the linear assignment problem and joint matrix diagonalization and the part of the computational cost that depends on the number of variables is at worst cubic. We have shown sufficient and necessary conditions under which the network is fully identifiable, which require observations from at least three different environments. The strength and location of interventions can also be reconstructed.

## Footnotes

[1]The blocking of all but one cycle can be achieved by do-interventions on appropriate variables under the following condition: for every pair of cycles in the graph, the variables in one cycle cannot be a subset of the variables in the other cycle. Otherwise the blocking could be achieved by deletion of appropriate edges.

[2]An R-package called "`backShift`" is available from CRAN.

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
