[Supplementary Material]



# BACKSHIFT: Learning causal cyclic graphs from unknown shift interventions

## A Identifiability – Proof of Theorem 1

*Proof.* "if": Let $\mathbf{D}'$ be a solution of (10). Let us write $\mathbf{D}'_{m\bullet}$ for the $m$-th row of $\mathbf{D}'$ and $\mathbf{D}_{m\bullet}$ for the $m$-th row of $\mathbf{D}$, $m = 1, \ldots, p$. Furthermore let us define $\mathbf{g}_m := \mathbf{D}^{-T}\mathbf{D}'_{m\bullet}$, $m = 1, \ldots, p$. We will show that at most one entry of this vector is nonzero. Note that by equation (7) we have $\mathbf{\Delta\Sigma}_{\mathbf{x},j} = \mathbf{D}^{-1}\mathbf{\Delta\Sigma}_{\mathbf{c},j}\mathbf{D}^{-T}$ for all $j \in \mathcal{J}$. By equation (7), $L(\mathbf{D}\mathbf{\Delta\Sigma}_{\mathbf{x},j}\mathbf{D}^T) = 0$. As $\mathbf{D}'$ solves equation (10), this implies $L(\mathbf{D}'\mathbf{\Delta\Sigma}_{\mathbf{x},j}\mathbf{D}'^T) = 0$ for all $j \in \mathcal{J}$. Hence the offdiagonal elements of $\mathbf{D}'\mathbf{\Delta\Sigma}_{\mathbf{x},j}\mathbf{D}'^T$ are zero, which implies

$$\mathbf{g}_{m'} \perp \mathbf{\Delta\Sigma}_{\mathbf{c},j}\mathbf{g}_m \text{ for all } m' \neq m \text{ and for all } j \in \mathcal{J}.$$

As the $\mathbf{g}_{m'}$ are linearly independent, this implies that for all pairs $j, j' \in \mathcal{J}$, $\mathbf{\Delta\Sigma}_{\mathbf{c},j}\mathbf{g}_m$ and $\mathbf{\Delta\Sigma}_{\mathbf{c},j'}\mathbf{g}_m$ are collinear i.e. for all $(j, j')$ there exists a $\lambda_{j,j'} \in \mathbb{R}$ such that $\mathbf{\Delta\Sigma}_{\mathbf{c},j}\mathbf{g}_m = \lambda_{j,j'}\mathbf{\Delta\Sigma}_{\mathbf{c},j'}\mathbf{g}_m$ or $\lambda_{j,j'}\mathbf{\Delta\Sigma}_{\mathbf{c},j}\mathbf{g}_m = \mathbf{\Delta\Sigma}_{\mathbf{c},j'}\mathbf{g}_m$

Take arbitrary $k, l \in \{1, \ldots, p\}$ and choose $j, j' \in \mathcal{J}$ such that (13) is satisfied. By the argumentation above, there exists a $\lambda_{j,j'} \in \mathbb{R}$ such that $\mathbf{\Delta\Sigma}_{\mathbf{c},j}\mathbf{g}_m = \lambda_{j,j'}\mathbf{\Delta\Sigma}_{\mathbf{c},j'}\mathbf{g}_m$ or $\lambda_{j,j'}\mathbf{\Delta\Sigma}_{\mathbf{c},j}\mathbf{g}_m = \mathbf{\Delta\Sigma}_{\mathbf{c},j'}\mathbf{g}_m$. Without loss of generality let us assume the latter. Recall that both $\mathbf{\Delta\Sigma}_{\mathbf{c},j}$ and $\mathbf{\Delta\Sigma}_{\mathbf{c},j'}$ are diagonal matrices. Now condition (13) implies that the $k$-th or the $l$-th entry on the diagonal of $\lambda_{j,j'}\,\mathbf{\Delta\Sigma}_{\mathbf{c},j} - \mathbf{\Delta\Sigma}_{\mathbf{c},j'}$ is nonzero (or both). Hence, the $k$-th or the $l$-th entry of $\mathbf{g}_m$ s zero (or both). By repeating this argumentation for all $k$ and $l$, at most one entry of $\mathbf{g}_m$ is nonzero. Thus, $\mathbf{D}'_{m\bullet} = \mathbf{D}^T\mathbf{g}_m = (\mathbf{g}_m^T\mathbf{D})^T$ is a multiple of one of the rows of $\mathbf{D}$.

By applying this argumentation for all $m = 1, \ldots, p$, each row of $\mathbf{D}'$ is a multiple of one of the rows of $\mathbf{D}$. As both $\mathbf{D}$ and $\mathbf{D}'$ are invertible, there exists a bijection between the rows of $\mathbf{D}'$ and $\mathbf{D}$ such that the corresponding rows are collinear. Furthermore, the diagonal of $\mathbf{D}'$ and $\mathbf{D}$ is $(1, \ldots, 1)$. Hence let us consider a bijection $\sigma : \{1, \ldots, p\} \mapsto \{1, \ldots, p\}$ such that the $\sigma(m)$-th row of $\mathbf{D}'$ is a multiple of the $m$-th row of $\mathbf{D}$, i.e. $\frac{1}{\mathbf{D}'_{\sigma(m),m}}\mathbf{D}'_{\sigma(m)\bullet} = \mathbf{D}_{m\bullet}$ for all $m = 1, \ldots, p$. We want to show that this bijection is the identity. First observe that, as the diagonal of $\mathbf{D}'$ and $\mathbf{D}$ is $(1, \ldots, 1)$, $\frac{1}{\mathbf{D}'_{\sigma(m),m}} = \mathbf{D}_{m,\sigma(m)}$ for all $m = 1, \ldots, p$. Now let us consider a cycle in this permutation , i.e. $m_1, \ldots, m_{\eta+1} = m_1, \eta > 1, m_\iota \neq m_\kappa$ for $1 \leq \iota < \kappa \leq \eta$ and with $\sigma(m_\iota) = m_{\iota+1}$ for $1 \leq \iota \leq \eta$. If this leads to a contradiction, we can conclude that $\sigma$ is the identity. As $\mathbf{D}_{m,m} = 1$, $\mathbf{D}'_{\sigma(m),m} \neq 0$, i.e. $\mathbf{D}'_{m_{\iota+1},m_\iota} \neq 0$ for $1 \leq \iota \leq \eta$. This corresponds to a cycle in with product

$$\prod_{\iota=1,\ldots,\eta} \mathbf{D}'_{m_{\iota+1},m_\iota} = \prod_{\iota=1,\ldots,\eta} \frac{1}{\mathbf{D}_{m_\iota,m_{\iota+1}}}. \tag{15}$$

As $\mathbf{D}'$ is a solution of (10), $CP(\mathbf{I}-\mathbf{D}') < 1$, hence the product on the left hand side of equation (15) is in absolute value strictly smaller than 1, see (2). Analogously, as $\mathbf{D}_{m_\iota,m_{\iota+1}} \neq 0$ for $\iota = 1, \ldots, \eta$, the sequence $m_{\eta+1}, m_\eta, \ldots, m_1$ corresponds to a cycle with product

$$\prod_{\iota=1,\ldots,\eta} \mathbf{D}_{m_\iota,m_{\iota+1}}.$$

Using the same argumentation as for $\mathbf{D}'$, this product is in absolute value strictly smaller than 1, which contradicts (15). Hence such cycles of length $\geq 2$ do not exist and $\sigma$ is the identity. Hence, $\mathbf{D}' = \mathbf{D}$.

"only if": As above define $\mathbf{D}_{m\bullet}$ as the $m$-th row of $\mathbf{D}$ and let us write $\mathbf{u}_m \in \mathbb{R}^p$ for the $m$-th unit vector for $m = 1, \ldots, p$. Assume that (13) is not true, i.e. there exist $k, l \in \{1, \ldots, p\}$ such that for all $j, j' \in \mathcal{J}$,

$$(\mathbf{\Delta\Sigma}_{\mathbf{c},j})_{kk}(\mathbf{\Delta\Sigma}_{\mathbf{c},j'})_{ll} = (\mathbf{\Delta\Sigma}_{\mathbf{c},j})_{ll}(\mathbf{\Delta\Sigma}_{\mathbf{c},j'})_{kk}. \tag{16}$$

Without loss of generality let us fix a $j' \in \mathcal{J}$ with $(\mathbf{\Delta\Sigma}_{\mathbf{c},j'})_{kk} \neq 0$, and define $\lambda := (\mathbf{\Delta\Sigma}_{\mathbf{c},j'})_{ll}/(\mathbf{\Delta\Sigma}_{\mathbf{c},j'})_{kk}$. If such a $j'$ does not exist, we can apply the same argumentation as below but with the $k$ and $l$ interchanged and $\lambda := 0$.

Note that the definition of $\lambda$ does not depend on $j$ and that by equation (7) we have $\boldsymbol{\Delta\Sigma}_{\mathbf{x},j} = \mathbf{D}^{-1}\boldsymbol{\Delta\Sigma}_{\mathbf{c},j}\mathbf{D}^{-T}$. Then, for $\delta \in \mathbb{R}$ we can define $\mathbf{D}'_{k\bullet} := \mathbf{D}_{k\bullet} + \delta\mathbf{D}_{l\bullet}$ and $\mathbf{D}'_{l\bullet} := \mathbf{D}_{l\bullet} - \delta\lambda\mathbf{D}_{k\bullet}$ and we obtain for all $j \in \mathcal{J}$

$$\begin{aligned}\mathbf{D}'^T_{l\bullet}\boldsymbol{\Delta\Sigma}_{\mathbf{x},j}\mathbf{D}'_{k\bullet} &= (\mathbf{u}_l - \delta\lambda\mathbf{u}_k)^T\boldsymbol{\Delta\Sigma}_{\mathbf{c},j}(\mathbf{u}_k + \delta\mathbf{u}_l)\\ &= \delta(\boldsymbol{\Delta\Sigma}_{\mathbf{c},j})_{ll} - \delta\lambda(\boldsymbol{\Delta\Sigma}_{\mathbf{c},j})_{kk}\\ &= 0.\end{aligned}$$

In the second equation we used (16). Furthermore, for small $\delta$ let us scale $\mathbf{D}'_{k\bullet}$ such that the $k$-th component of the vector is 1. Analogously, let us scale $\mathbf{D}'_{l\bullet}$ such that the $l$-th component of the vector is 1. Then we can define the matrix $\mathbf{D}'$ as the rows of $\mathbf{D}$ except for row $k$ and $l$ which are replaced by $\mathbf{D}'_{k\bullet}$ and $\mathbf{D}'_{l\bullet}$. By above reasoning, this matrix satisfies

$$\mathbf{D}'\boldsymbol{\Delta\Sigma}_{\mathbf{x},j}\mathbf{D}'^T \in \mathrm{Diag}(p)$$

for all $j \in \mathcal{J}$ and $\mathbf{D}'$ is invertible. Furthermore, the diagonal elements of $\mathbf{D}'$ are 1. Recall that the path-products of $\mathbf{I} - \mathbf{D}$ over cycles are in absolute value smaller than 1, see (2). For small $\delta$, $\mathbf{I} - \mathbf{D}'$ is close to $\mathbf{I} - \mathbf{D}$ (in an arbitrary matrix norm) and hence the path products of $\mathbf{I} - \mathbf{D}'$ over cycles are in absolute value smaller than 1 as well. As $\mathbf{D}$ is invertible, $\mathbf{D}' \neq \mathbf{D}$. Hence the solution to (10) is not unique. This concludes the proof.

$\square$

## B Polynomial-time algorithm

Here, we provide the necessary theoretical result to show that BACKSHIFT has a computational cost of $O(|\mathcal{J}| \cdot n \cdot p^2)$. Specifically, we show that Step 3 in Algorithm 1 can be cast in terms of the classical linear sum assignment problem, having a computational complexity of $O(p^3)$.

**Theorem 3.** *Let $\mathbf{D} \in \mathbb{R}^{p\times p}$ be a matrix with $CP(\mathbf{D}) < 1$, $diag(\mathbf{D}) \equiv 1$ and $\mathbf{D}_{k,l} \neq 0$ for $k, l \in \{1,\ldots,p\}$. For $\mathbf{D}' \in \mathbb{R}^{p\times p}$ define*

$$P(\mathbf{D}') := \prod_{k,l}|\mathbf{D}'_{k,l}|.$$

*Furthermore define*

$$\mathcal{D}_p := \{\mathbf{D}' : \textit{There exists a permutation } \sigma \textit{ of } \{1,\ldots,p\} \textit{ such that the } \sigma(m)\textit{-th row of } \mathbf{D}$$
$$\textit{is collinear to the } m\textit{-th row of } \mathbf{D}' \textit{ and } diag(\mathbf{D}') \equiv 1 \}.$$

*Then,*

$$\mathbf{D} = \arg\min_{\mathbf{D}'\in\mathcal{D}_p} P(\mathbf{D}') = \arg\min_{\mathbf{D}'\in\mathcal{D}_p} \log P(\mathbf{D}').$$

*Proof.* Let $\mathbf{D}' \in \mathcal{D}_p$ with $\mathbf{D}' \neq \mathbf{D}$. Let us write $\mathbf{D}_{m\bullet}$ for the $m$-th row of $\mathbf{D}$ and analogously $\mathbf{D}'_{m\bullet}$ for the $m$-th row of $\mathbf{D}'$, $m = 1, \ldots, p$. Now let $\sigma$ be a permutation such that the $\sigma(m)$-th row of $\mathbf{D}$ is collinear to the $m$-th row of $\mathbf{D}'$. As $\mathbf{D}' \neq \mathbf{D}$, we have that $\sigma \neq \mathrm{Id}$. As $diag(\mathbf{D}') \equiv 1$,

$$\frac{1}{\mathbf{D}_{\sigma(m),m}}\mathbf{D}_{\sigma(m)\bullet} = \mathbf{D}'_{m\bullet}.$$

It immediately follows that

$$\left(\prod_{m=1,\ldots,p}\frac{1}{|\mathbf{D}_{\sigma(m),m}|}\right)^p P(\mathbf{D}) = P(\mathbf{D}').$$

As $CP(\mathbf{D}) < 1$ and $\sigma$ is not the identity, $\prod_{m=1,\ldots,p}|\mathbf{D}_{\sigma(m),m}| < 1$. As all elements of $\mathbf{D}$ and $\mathbf{D}'$ are nonzero, $P(\mathbf{D}) > 0$ and $P(\mathbf{D}') > 0$. Hence, $P(\mathbf{D}') > P(\mathbf{D})$. This concludes the proof.

$\square$

**Remark:** We can define the relative loss function of moving row $k$ to row $l$ as

$$\ell(k, l) = -\log(|\mathbf{D}'_{k,l}|).$$

Then the linear assignment problem that minimizes this problem also yields the correct permutation for Step 3 in Algorithm 1 if it exists, i.e. the permutation $\sigma$ on $\{1, \ldots, p\}$ that minimizes

$$\sum_{k=1}^{p} \ell(k, \sigma(k))$$

satisfies that $\mathbf{D}'_{m\bullet}$ is collinear to $\mathbf{D}_{\sigma(m)\bullet}$.

**Remark:** Allowing for self-loops would lead to an identifiability problem, independent of the method. For every model with self-loops and $CP < 1$ there is a model without self-loops and $CP \leq 1$ yielding the same observational distribution in equilibrium. The connectivity matrix without self-loops can thus be seen as a representative of a whole class of connectivity matrices that allow self-loops. Specifically, if the connectivity matrix with self-loops is $\mathbf{B}^*$, define matrix $\mathbf{T}$ by $\texttt{PermuteAndScale}(\mathbf{I} - \mathbf{B}^*) = \mathbf{T}(\mathbf{I} - \mathbf{B}^*)$, where $\texttt{PermuteAndScale}()$ is the operation defined in Step 3 of the BACKSHIFT algorithm. Technically, $\texttt{PermuteAndScale}()$ is only defined for matrices that are nonzero outside of the diagonal. Using similar arguments as in Theorem 3, $\texttt{PermuteAndScale}()$ can be extended to arbitrary matrices with nonzero diagonal elements. To be more precise, there exists a matrix $\mathbf{T}$ such that $CP(\mathbf{T}(\mathbf{I} - \mathbf{B}^*)) \leq 1$, $\mathrm{diag}(\mathbf{T}(\mathbf{I} - \mathbf{B}^*)) \equiv 1$ and such that $\mathbf{T}$ is the product of a diagonal scaling matrix with a permutation matrix. Then define $\mathbf{B}_{new} := \mathbf{I} - \mathbf{T}(\mathbf{I} - \mathbf{B}^*)$, $\mathbf{e}_{j,new} = \mathbf{T}\mathbf{e}_j$ and $\mathbf{c}_{j,new} = \mathbf{T}\mathbf{c}_j$ for all $j \in \mathcal{J}$. As $\mathbf{T}$ is the product of a diagonal scaling matrix with a permutation matrix, assumptions (B) and (C) are still fulfilled and $\mathbf{x}_{j,new} = (\mathbf{I} - \mathbf{B}_{new})^{-1}(\mathbf{e}_{j,new} + \mathbf{c}_{j,new}) = (\mathbf{I} - \mathbf{B}^*)^{-1}(\mathbf{e}_j + \mathbf{c}_j) = \mathbf{x}_j$ for all $j \in \mathcal{J}$. This implies that the two matrices $\mathbf{B}^*$ with self-loops and $\mathbf{B}_{new}$ without self-loops (since it has zeroes on the diagonal by construction) have both $CP \leq 1$ and yield the same distribution.

## C   Intervention variances and model misspecification

The method allows to validate and check the assumptions to some extent. This is especially important in the data of [22] as pointed out in [26]. The interventions can mostly be thought of as not changing the concentration of a biochemical agent but rather changing the activity of the agent, for example by inhibiting the reactions in which the agent is involved [26]. Under such a mechanism change, it is doubtful whether the interventions are well approximated by our model (3) with independent shift-interventions. We can check the assumptions by the success of the joint diagonalization procedure. Specifically, we get an empirical version of (7) when plugging in the estimators and can check whether all off-diagonal elements on the right hand side of (7) are small or vanishing. We list below results for the seven experimental intervention conditions whose target is well described in [26]. The element on the right-hand side of (7) with the largest absolute value is selected. We use now the Gram instead of the covariance matrix to be also sensitive to model-violations of the additional assumption (C'), see Section D, though the results are almost identical whether using the Gram or covariance matrix. These large off-diagonal elements indicate a violated mechanism in the sense that the model (3) does not fit very well, because either the interventions have not been of the assumed shift-type or the causal mechanism in which the agent is involved has changed under the intervention.

| Experiment | Reagent | Intervention | largest mechanism violation |
|---|---|---|---|
| 3 | Akt-Inhibitor | inhibits AKT activity | PLCg $\leftrightarrow$ PKA |
| 4 | G0076 | inhibits **PKC** activity | **PKC** $\leftrightarrow$ PIP2 |
| 5 | Psitectorigenin | inhibits **PIP2** abundance | **PIP2** $\leftrightarrow$ PKA |
| 6 | U0126 | inhibits **MEK** activity | **MEK** $\leftrightarrow$ PKA |
| 7 | LY294002 | changes PIP2/PIP3 mechanisms | PKA $\leftrightarrow$ JNK |
| 8 | PMA | activates PKC activity | MEK $\leftrightarrow$ PKA |
| 9 | $\beta$2CAMP | activates **PKA** activity | **PKA** $\leftrightarrow$ PKC |

The table above lists the results for the seven experimental conditions where we know the intervention mechanism, at least approximately. The results are interesting in that the most violated

mechanism (the largest entry in the off-diagonal matrix on the right-hand side of the empirical version of (7)) occurs in 4 of the 7 experimental conditions directly at the intervention target. In 3 of these 4 cases, the violated mechanism concerns a relation that has a large entry in the estimated connectivity matrix. This corresponds well with the model of activity interventions in [26]. Note that we have not made use of the intervention targets in the estimation procedure. The interesting point is that we can use the model violations to estimate with some success where the interventions occurred.

## D  Beyond covariances

For the method above, we exploit differences in the covariance of observations across different environments. We can also exploit a shift in the mean of the intervention strength $\mathbf{c}$ (and consequently in the observations $\mathbf{x}$) when strengthening the condition (C) to (C'). Specifically, we require for (C') that in each environment $j \in \mathcal{J}$ the shift in the mean $E(\mathbf{c}_j)$ equals zero for all variables except at most one variable. The variable with a non-zero shift in the mean can change from one environment to another. Note that the counterpart of (5) when using the Gram matrix instead of the covariance matrix reads

$$(\mathbf{I} - \mathbf{B})\mathbf{G}_{\mathbf{x},j}(\mathbf{I} - \mathbf{B})^T = \mathbf{G}_{\mathbf{c},j} + \mathbf{G}_{\mathbf{e}}. \tag{17}$$

Under the stronger version (C'), the difference across environments of the right-hand side in (17) is again a diagonal matrix and we can proceed just as above, by replacing the covariance matrices with Gram matrices throughout. If the assumption (C') is satisfied, this allows identifiability of the graph in a wider range of settings (Theorem 1 can be adapted in a straightforward manner by again replacing covariances with Gram matrices) but requires the stricter condition (C'). Since in practice it is often unclear whether the stricter condition is approximately true, we work mainly with the weaker assumption (C) and exploit only shifts in the covariance matrices.

## E  Additional figures

Figure 5: Synthetic data. Stability selection results for BACKSHIFT with parameters $\mathbb{E}(V) = 2$ and $\pi_{thr} = 0.75$. The intensity of the edges illustrates the relative magnitude of the estimated coefficients, the width shows how often an edge was selected. The edge from node 6 to node 10 is associated with the smallest coefficient in absolute value. It is retained in none of the settings in the stability selection procedure.