[Reviews · NeurIPS 2015]

Submitted by Assigned_Reviewer_1

This is an original method for uncovering causal relationships in linear cyclic model with uncertain interventions. The method is both fairly original and relatively simple, also providing an intuitive sense of which experimental setups are necessary for identifiability (I have not read the Appendix, but Theorem 1 is intuitive). This problem is hard, and although it is easy to criticize the method concerning possibly strong assumptions and the possibly high-variance estimators used (not to mention the possible lack of robustness to small misspecifications), the matter of fact is that this seems to be more general than current approaches, with a simple and elegant algorithm.

In one sense, the premises for Theorem 1 are also assumptions of the model. Were they checked for the case studies of Section 4.2 and 4.3?

How realistic is assumption B? I understand why it is helpful, and by all means it is a good starting point for a new method (which already has fewer assumptions than existing ones), but if you expect fat hands on the observables, why not on the hidden common causes too?

In reality, one has quite some information about the direct effects of the external shocks. Arguably it would be more useful to allow for changes in some hidden common causes (in some way), but where we can exclude some of the candidate observable locations a priori.

In Line 343, I do not understand the comment that thse data points cannot ``be regarded as independent anymore''. Based on the given description, I have no idea why this is the case.

The experiments with synthetic data provide convincing evidence of advantages of the new method compared to Ling, but I would be happier that if instead of considering 0/1 losses for each edge, we had some continuous error measure such as the mean squared error of \hat{B}.

The analysis in 4.3 seems potentially very interesting, but again I think it could be better explained. What are the intervention scenarios? Are Figures 4(c) and (d) separate independent estimates for each of the 74 blocks? Are the assumptions believable for this domain?

Summary: A very general approach for learning causal linear cyclic models with uncertain interventions. Intuitive identifiability results and an elegant algorithm are provided.

Submitted by Assigned_Reviewer_2

Sorry but I will not be able to review this paper.
Summary: Sorry but I will not be able to review this paper.

Submitted by Assigned_Reviewer_3

%%%%%%%%%%%%%%%%%%%%%%%%%%% Update after authors response: I thank the authors for taking the time to clearly and honestly respond to my (our) questions. I am even more convinced that the work is worth presenting at NIPS. Probably my lack of understanding (or me being too lazy to read the appendices) but would really like to see the step-by-step explanation on the link between Assumption C and the ability of the algorithm to identify situations where the permutation (1 step only? Other remark: possible since you only have few nodes as well right otherwise the permutation space simply can't be explored?) which makes CP < 1.

%%%%%%%%%%%%%%%%%%%%%%%%%%% The manuscript introduces linear causal dependencies in a complex system as measured by x's. The originality is that the location of the intervened variables is unknown. The authors present assumptions (section 2.3) under which they derive theoretical results in terms of model identifiability (section 3), in both an ideal setting where the sample size is infinite (algorithm presented in section 2.4) and when it isn't (algorithm presented in section 2.5). A good aprt of the paper is devoted to much appreciated numerical experiments on both simulated (ground truth known) and real data networks.

I quite appreciated reading this paper and found the problem well motivated and nicely tackled. The only nuance is that it is not a huge breakthrough, rather a nice and useful increment from existing theory.

Comments to the authors:

- l36, def of "equilibrium data" is never clear.

- l44, "solutions to (1)": mind that (1) is only a model. I guess you mean solutions of an optimisation problem related to Eqn (1) by a score function; this is detailed in sections 2.4 and 2.5 I know so merely a language issue here.

- l61: "surgical" used here and defined later.

- l62: iterations? Haven't talked about an algorithm yet

- l64: interactionS (typo)

- l84: unknown location of interventions; OK why not but in fields like modern Molecular Biology, this is fully determined, e.g. gene KO. Or on the contrary there would be way too many interventions at the same time (e.g. genetical genomics data sets where tiny interventions are performed at each node at the same time, see Mooij et al NIPS 2015 or Vignes et al. PLoS ONE 2011)

- l102-103 is intriguing, could you clarify?

- section 2.3: the 3 assumptions are in fact 3 (coherent?) sets of assumptions? Sentences in each item are not always equivalent. Moreover I-B invertible roughly forbids considering high-dimensional or even moderate dimension scenarii. So you don't really compare to many approaches you quote? With nice implementations like Kalisch and Buhlmann JMLR 2007 or their pcalg R package. In the same vein, see also Rau et al. BMC Syst. Biol. 2013.

- l177: can you expand why Eqn (7) motivates the estimator in Eqn (10)? Rational for the loss defined by L as well?

- l204, can you quickly explain how FFDiag [23] ( I mean how it works, reads like the difficult part in your computational work) allows you to solve problem (11)?

- l207: (a) permuting -> looks like a permutations of potentially many edges? Can be harsh at the end or do I understand incorrectly? For permutation effect on columns of an adjacency matrix, see Champion et al. http://arxiv.org/abs/1507.02018

- l212, variant of LAP -> any guarantee of an acceptable solution at this step?

- l268: producing the code is VERY appreciated!
Summary: This work is about the reconstruction of causal network. The article is well presented and the results are worth being presentend in my opinion.

Submitted by Assigned_Reviewer_4

This paper proposes an algorithm for learning linear causal models which may be both cyclic and contain latent variables from observational and experimental conditions, but does not require that inverventions be surgical or known, only that interventions on at different variables in the same experimental setting be uncorrelated. To my knowledge, this is a much weaker condition than competing approaches. The authors' prove an identifiability condition for their approach: data from at least 3 different experimental settings (one of which may be observational) are needed. Their method seems to perform well on observational data and flow cytometry data.

The paper is clearly written and well organized in general. The technical work appears to be sound. The approach is well motivated as it lends itself to the type of data flow cytometry produces. One way this paper could be improved would be for the authors to give a more thorough background on existing methods for learning causal models from experimental data under various assumptions (cyclicity, latent variables, types of interventions, etc.) and where their approach fits in and how it improves over these approaches. Also, the financial time series application is not very clear and the presentation is confusing (though I think this is a good paper without this application).
Summary: This paper proposes an algorithm for learning linear causal models which may be both cyclic and contain latent variables from observational and experimental conditions, but does not require that interventions be surgical or known, proves that the full model is identifiable if 3 experimental settings are available, and applies the model to synthetic data, flow cytometry data, and financial time series data. The paper is generally well written, sound, well motivated by applications, and seems to perform well and require fewer assumptions than existing approaches.

Author Feedback
Author rebuttal: We thank the reviewers for their positive feedback, insightful comments and helpful suggestions. We wish to make the following clarifications.

Assumptions (R1,R3,R4)

Assumption A (R4). backShift is not applicable in the high-dimensional setting in its current form. This would require some form of suitable sparsity assumption on the network (similar to sparsity assumptions on the coefficients in high-dim. regression) which could be investigated in future work.

Assumption B (R1). Additionally allowing for interventions on the latents would be an interesting extension. As you say, one would then have to use some more knowledge about the location of the interventions.

Assumption C (R1,R3,R4). It is indeed a limitation that the interventions on different variables have to be uncorrelated but it allows us to make very few assumptions on the underlying causal structure. Notably, we can detect a violation of Assumption (C) by the success or failure of the joint diagonalization procedure: Assume the population case and that at least three environments fulfill the conditions of Theorem 1 but adding the remaining environments would violate them and in particular Assumption (C). Take any three environments that do satisfy the conditions and they will already uniquely determine the network B that leads to a joint diagonalization of the difference in covariance matrices. Any additional environment that violates the uncorrelatedness will then lead to a situation where joint diagonalization is impossible. This was checked for the case studies in Sec. 4.2 (see Appendix C) and 4.3 (diagonalization succeeded). We aim to formalize this in future work. (@R4: l.399-406 and Appendix C elaborate on the statement made in l.102-103.)

"No self-loops" (R3): Allowing for self-loops would lead to an identifiability problem, independent of the method under consideration. We will clarify this further in the manuscript.

Clarifications (R1, R4, R5)

a) Intervention type (R4)
backShift *can* be used when there are many interventions in one experiment. It is not applicable if the interventions are not shift-type but e.g. surgical instead (as in gene knock-outs). The joint diagonalization (cf. Eq.(7)) will be impossible if the interventions are not all of a shift-type nature, for analogous reasons as in the previously discussed case of a violation of Assumption (C).

b) Algorithm guarantees (R4)
We define a loss function so that the LAP minimizes the cycle product (see Appendix B; cf. l.596). At most one permutation can lead to CP<1 (cf. Thm 3) if such a permutation exists. If it does exist, our algorithm finds it. If, on the other hand, no permutation with CP<1 exists, we detect this as well (i.e. a connectivity matrix with CP>=1 is returned) in which case the model assumptions are not met. The permutation step is solved by this variant of the LAP and is thus computable in O(p^3).

c) Financial time series (R1,R5)
The financial time series in Sec. 4.3 is an instance where there are no explicitly controlled interventions but the external input is changing over time. Therefore, we group the data into 74 overlapping blocks of consecutive observations so that each of these blocks corresponds to one environment. We then estimate the connectivity matrix B with backShift and LING. These estimates are plugged into (12), respectively, yielding the estimated intervention variances for each of the three stock indices and in each of the 74 environments (modulo an offset; also see Section 2.6), shown in Fig.4(c) (backShift) and 4(d) (LING): The intervention variances estimated using the backShift estimator separate the origins of the three major downturns very clearly. On the other hand, using the LING estimator does not yield such a clear separation.

d) Misc (R1,R4)
R4: l.177: Eq.(7) motivates the estimator in Eq.(10) because the rhs of Eq.(7) is a diagonal matrix for all settings j (cf. the definition in (6)). We thus need to find the joint diagonalizer I-B. This is the rationale for the loss function: L(A)=0 iff A is a diagonal matrix.

R1: l.343: We will clarify this statement in the paper.